# ^13^C, ^25^Mg, and ^43^Ca Solid-State NMR for the Purpose of Dolomitic Marbles Provenance Elucidation

**DOI:** 10.3390/ma16041468

**Published:** 2023-02-09

**Authors:** Isabelle Pianet, Anna Gutiérrez Garcia-Moreno, Marie-Claire Savin, Nicolas Frerebeau, Julien Trebosc, Pierre Florian, M. Pilar Lapuente Mercadal

**Affiliations:** 1Department of Archaeology, Archéosciences Bordeaux, UMR CNRS-Université Bordeaux-Montaigne 6034, Maison de l’archéologie, Esplanade des Antilles, 33600 Pessac, France; 2Unitat d’Estudis Arqueomètrics, Institut Català d’Arqueologia Clàssica, Plaça d’en Rovellat, 43003 Tarragona, Spain; 3Departament de Ciències de l’Antiguitat i de l’Edat Mitjana, Universitat Autònoma de Barcelona, Cerdanyola del Vallès, 08193 Barcelona, Spain; 4Department of Chemistry, Institut Michel Eugène Chevreul, CNRS FR 2638, Université Lille, Avenue Paul Langevin, 59655 Villeneuve d’Ascq, France; 5Department of Chemistry, Conditions Extrêmes et Matériaux: Haute Température et Irradiation UPR CNRS 3079, 1D Avenue de la Recherche Scientifique, 45071 Orléans, France; 6Department of Earth Sciences, Petrology and Geochemistry, University of Zaragoza (UNIZAR), C/Pedro Cerbuna, 12, 50009 Zaragoza, Spain

**Keywords:** marble, dolomite, elemental analysis, principal component analysis, NMR, provenance

## Abstract

The study of the provenance of dolomitic marble artefacts has become relevant since it was discovered that quarries of this marble other than that of Cape-Vathy located on the island of Thasos have been exploited since Antiquity. To improve our knowledge about the provenance of materials and the extent of their dispersion, multiple archaeometric studies were performed in the past including isotope analyses, petrography, cathodoluminescence, and elemental analyses. In the present work, solid-state nuclear magnetic resonance (NMR) spectroscopy has been added to this panel of techniques. NMR allows the characterization of the material at a molecular level by looking at different nuclei: carbon, magnesium, and calcium. Statistical analysis of the data collected on both quarry samples and archaeologic items was also implemented and clearly demonstrates the efficiency of a holistic approach for provenance elucidation. Finally, the first ^25^Mg NMR tests have shown the potential of this technique to discriminate between dolomitic marbles of different provenance. The results are discussed in terms of their historical meaning and illustrate the exploitation of sources of dolomitic marbles other than the Greek Thasos source.

## 1. Introduction

The characteristics of dolomitic marbles have been described since 1791 by the Commander Déodat de Dolomieu, a French geologist, in a letter addressed to Mr. Picot de La Peyrouse and published in the Journal de Physique [1]. This letter, titled “On a type of limestone that is not very effervescent with acids, and phosphorescent on collision”, refers to what the artists called “*marmo graeco duro*”, frequently used in Ancient Rome to sculpt colossal statues and that reacts differently with respect to acids than calcitic marbles. During his escapades in the Tyrolean mountains, he also discovered limestone with the same physical properties which was named “Dolomiti” in honor of Dolomieu by his colleague Theodore de Saussure [2].

Since that time, the two varieties of white marbles, dolomitic and calcitic, have been studied and can be distinguished by their chemical composition: calcite is a crystalized calcium carbonate, whereas dolomite is a crystalized calcium magnesium carbonate. The two marbles were widely used in the ancient world, particularly during the imperial age in architecture and plastic arts, and the dolomitic marble was specifically chosen for fine arts because of its high quality and its relative hardness compared with calcitic marbles.

Still today, many archaeologists attribute the provenance of an ancient artifact made of a coarse grain and a very uniform white dolomitic marble to the Cape-Vathy quarry of the Thasos Island [3,4,5,6]. While rather scarce, its presence is not limited to the Thasos Island in the Mediterranean Basin. Indeed, the existence of other quarries, probably exploited since the Late Antiquity, has been revealed: this is the case of the quarries of the Sivec mountains in Macedonia [7]; Crevola in Italy [8]; Mijas-Coín located in the Málaga province, in the Hispanic Baetica roman province [9]; or those of the quarry of Lez in the French district of Lez-Saint-Béat, Haute Garonne [10], whose use for archaeologic artifacts is being investigated. To the extent of our knowledge, the distribution of these marbles during Roman times followed different routes: the Greek marble is found throughout much of the Roman Empire, whereas the other marbles from Spain, Italy, France, or Macedonia appear to be limited to local use.

Two strategies may be employed to elucidate marble provenance. The most commonly used strategy consists of comparing scientific data acquired for an archaeological item to those obtained from different quarries by different teams and reported in scientific publications. This strategy, which involves the elimination of certain sources at each stage, can be called a “phase-out” strategy. Two major methods are employed for this comparison: (i) the mineralogical and petrographic analysis of thin sections and (ii) the analysis of the stable carbon and oxygen isotope ratio. All the reference databases have been published for most of quarries known to be exploited during Antiquity: Greece, Italy [11], Spain [12], Portugal [13], France [14], and so on. To these two unavoidable methods, a third one has been added by some authors such as cathodoluminescence [15], fluid inclusion [16], elemental analyses [17], and electronic paramagnetic resonance [18].

The second strategy processes all the data together through principal component analysis to determine the marble source. For this purpose, it is necessary to collect all the measures on the same set of samples (quarries and artefacts). Therefore, to test this holistic approach, other techniques may be added to the panel of the complementary techniques usually used, without necessarily having a database on the techniques developed over several decades.

This is this global approach that we decided to test in the present work, by collecting data on both geologic and archaeologic samples. To the established methods used, such as isotopic and elemental analysis, we add a third technique, solid-state nuclear magnetic resonance (NMR), which allows the deciphering of materials at a molecular level. In preliminary works, ^13^C NMR has been successfully developed for determining the provenance of some white calcitic marbles at a region scale [19] or even at a quarry level [20]. In the present work, in addition to carbon, two other nuclei—magnesium (^25^Mg isotope) and calcium (^43^Ca isotope)—are investigated in order to track differences at the molecular scale between the dolomitic varieties that can be useful for their discrimination.

Finally, all the data collected were processed using statistical analysis—principal component analysis—and access was provided to the provenance of the dolomitic marble of some decorative architectural materials of the Roman Theatre of Zaragoza, as well as to some sculptural artefacts conserved in the archaeological Museum of Málaga.

## 2. Materials and Methods

### 2.1. Materials

A large geological collection of dolomitic marbles related to the artifacts in this study was established. The collection includes dolomitic marbles from the Cape-Vathy quarry of the Thasos Island (seven samples provided by T. Kozelj [21]); the south of Spain, province of Málaga (four from Coín and four from Mijas), already studied by Lapuente et al. [9]; and from the French Pyrénées (seven samples from the dolomitic-marble quarries from the municipality of Lez-Saint-Béat) [10].

In order to test the experimental approach, five archaeological items were studied. Three of them (6 MM, 26 MM, and 29 MM) were sampled from the Archaeological Museum of Málaga, where they are currently conserved. They were discovered in different places in the vicinity of Málaga and they have already been the subject of a provenance study [9]. Two of them are decorations (plate and moulding) of the Roman Theatre of Zaragoza, and have also been previously studied [22]. All of the archaeological items studied are reported in Appendix A and displayed in Figure 1.

### 2.2. Analytical Methods

#### 2.2.1. NMR

^13^C, ^43^Ca, and ^25^Mg solid-state NMR, whose physico-chemical characteristics are reported in Appendix A, were performed on different apparatus.

^13^C NMR spectra of all of the samples were recorded with a 9.4 T Avance NMR apparatus from Bruker (Wissembourg, France) equipped with a broad band solid-state CPMAS 4 mm probe containing up to 200 mg of marble powder (CESAMO facilities) using the experimental protocol previously described [19].

^43^Ca NMR spectra of one sample of each quarry and two archaeological items (29 MM and BL) were recorded with a 21.1 T Avance Neo (Bruker, Wissembourg, France) equipped with a 7 mm VTN HX low gamma probe whose rotor contains up to 600 mg of marble powder (IMEC-ISB-UCCS Lille facilities) using the following parameters, as described in [23]: spectral width of 412 ppm (25,000 Hz) at 60.6 MHz a 90° pulse of 18.9 µs, an interpulse delay of 7 s, 12,000 repetitions corresponding to an acquisition time of around 24 h, and a rotation rate of 4 KHz.

^25^Mg NMR spectra of one sample of each quarry and two archaeological items (29 MM and BL) were recorded on a 18.8 T Avance Neo (Bruker, Wissembourg, France) equipped with a 4 mm HX low gamma probe [24] using the following parameters as proposed in [25]: a spectral width of 20,400 ppm (1 MHz) at 48.98 MHz, a 90° pulse of 3.5 µs, an interpulse delay of 17 s, 2048 repetitions of corresponding to an acquisition time of 10 h, and a rotation speed of 10 kHz.

The spectra were simulated as follows: ^13^C line shapes were simulated with Bruker Biospin software (version 4.2.0, 2022, Bruker, Wissembourg, France). ^43^Ca line shapes were accounted for assuming a central-transition second-order quadrupolar broadening defined by a quadrupolar coupling constant C_Q_ and an asymmetry parameter η_Q_, on top of the position (isotropic chemical shift δ_iso_) and a convolution by a Gaussian broadening rendering a distribution of δ_iso_ using DMFIT software [26]. ^25^Mg spectra simulations were performed using an “extended Czjzek” model [27,28]. Apart from the line intensity, this model has six adjustable NMR parameters: isotropic chemical shift δ_iso_, distribution of isotropic chemical shift δ_iso_, average quadrupolar coupling constant C_Q_, average asymmetry parameter η_Q_, distribution of quadrupolar coupling constant ΔC_Q_, and “disorder weight” ε. For both quadrupolar nuclei, infinite spinning speed regime and perfect magic angle were assumed. Spinning sidebands of the external transitions (seen as additional small intensities on the left side of the main peak) are not taken into account. 

#### 2.2.2. Elemental Analysis

Approximately 100 mg of each of the quarry and archaeological samples was dissolved in a 10 mL solution containing 3 mL HNO_3_ 68%, 3 mL of HCl 38%, 3 mL of HF 40%, and 1 mL of HClO_4_ 72% (Sigma Aldrich, Saint Quentin Fallavier, France). The mixtures were heated for 5 h at 130 °C in a closed container, and then at 190 °C until the total evaporation of all of the acids, as previously described [19]. The remaining powders were dissolved in a 5% HNO_3_ solution before the ICP-OES analysis (720ES, Varian, Palo Alto, CA, USA, at the Institut de Chimie de la Matière Condensée de Bordeaux (ICMCB) facilities). Ten discriminant elements with respect to provenance were dosed—Al, B, Ba, Fe, K, Mg, Mn, Si, Sr, and Ti, as previously described [19]. The compositional data obtained by ICP-OES (expressed in ppm) were transformed prior to statistical analysis using the centered log ratio transformation [29].

#### 2.2.3. Statistical Software

All collected data were analyzed with R version 4.2.0 (2022-04-22,) [30] and the following packages: dimensio 0.2.2 [31], dplyr 1.0.9 [32], ggplot 2 3.3.6 [33], ggrepel 0.9.1 [34], khroma 1.9.0 [35], patchwork 1.1.1 [36], robCompositions 2.3.1 [37], and tidyr 1.2.0 [32].

## 3. Results

The main goal of this study is to determine the provenance of marbles used to sculpt the five different artefacts displayed in Figure 1 using a holistic approach using data collected on archaeological and geological samples from three different techniques—^13^C NMR, elemental analyses, and ^13^C/^18^O isotopic ratio (the ratio ^13^C/^18^O of most of the samples (19) studied in this work were already reported in previous works [9,10]). All of the data were processed using principal component analyses to identify the provenance of the marble used to carve the artifacts. On the other hand, ^43^Ca and ^25^Mg NMR methods were tested to evaluate their potential for solving marbles’ provenance questions. The results are presented with respect to different purposes: developing analytical methods that are complementary to those that already exist, optimizing a confident provenance attribution strategy, and answering questions about dolomitic marble trade during Late Antiquity.

### 3.1. Solid-State NMR Investigation

First of all, it is important to note some peculiar specificity of the NMR experiments. Spectra were recorded on approximately 200 mg (for ^13^C and ^25^Mg) or 600 mg (for ^43^Ca) of powdered marble samples; the quantity depends on the size of the NMR sample holder used for the experiments. However, because NMR is a non-destructive method, the same powder used for all the NMR experiments was subsequently used for ICP-OES (100 mg) and isotopic analyses (maximum 30 mg).

^13^C NMR spectra (^13^C is the isotope that can be observed by NMR, see Appendix A) are sensitive to the presence of paramagnetic elements (mainly Fe) between the crystal lattice (^13^C spin-lattice relaxation rate modifies the signal intensity) and to the substitution of Ca or Mg by another element, leading to a broadening of the signal resonance [19]. These spectral characteristics have been successfully applied to elucidate calcitic marble provenance [20,38]. In the case of marble containing both calcite and dolomite phases, even if the ^13^C NMR chemical shifts of dolomite (170.8 ± 0.1 ppm) are slightly different than that of calcite (170.5 ± 0.1 ppm), the resolution limit is reached when the stone is composed of both phases. In this case, only one large resonance can be observed (Figure 2A). Nonetheless, if it is difficult to assign the ^13^C resonance peak to calcite or dolomite, the signal response (area and full width at half-maximum Δν_1/2_ in Hz) can be partially used as provenance markers. Table 1 reports on all the values: chemical shifts (close to 170.8 ppm) confirm that dolomite is the major component of all of the marbles analysed. However, differences are observed for both the resonances area and full width at half-maximum—Δν_1/2_—depending on the geographic location of the marble. Figure 3 displays the peak area against Δν_1/2_ for all data; this highlights the limit of the location determination using only ^13^C NMR data. Indeed, strong overlaps prevent the provenance identification of most artifacts, except for 29 MM, probably made of Coín marble, confirming the previous identification [9].

^43^Ca NMR spectra were successfully recorded despite the difficulties due to the low natural abundance of the isotope sensitive to NMR (0.135%, its quadrupolar nature, which broadens the signal, and the very low magnetogyric ratio of this isotope (Appendix A)). ^43^Ca chemical shift range (over 275 ppm [28]) allows to easily distinguish the dolomitic from calcitic marbles (Figure 2B): calcite resonates at 21.7 ppm (δ_iso_ 22.7 ppm) and the best fit for simulating its shape was obtained with a Gaussian–Lorentzian ratio of 200/80, a quadrupolar coupling constant C_Q_ of 1.4 MHz, and an asymmetry parameter η_Q_ of 0.1, values in agreement with previous reported results [29]; dolomite has a resonance centered at 11.7 ppm (δ_iso_ 14.0 ppm) and the best fit for simulating its shape was obtained with a C_Q_ value of 1.8 MHz and a η_Q_ value around 0.4. The ^43^Ca NMR chemical shift value observed for the dolomitic marble is close to the one previously observed for Mg-calcite [30], but to the best of our knowledge, this is the first time that the physico-chemical entities δ_iso_, C_Q_, and η_Q_ are reported for ^43^Ca spectra of dolomite.

As NMR spectra were recorded in quantitative mode (the delay between two pulses, 7 s, was long enough to ensure the total relaxation of the signal [23]), the areas of each resonance are proportional to the ratio of dolomite and calcite present in the stone, by taking into account the ratio 2:1 calcium atoms’ proportion with respect to one carbonate function of calcite or dolomite. In this sample, the calcite/dolomite ratio is estimated at 70:30.

While each ^43^Ca spectrum required a 24 h acquisition time, we chose to record three ^43^Ca NMR spectra of dolomitic marbles representative of each of the quarries, except for Mijas, and two artifacts, one from the Roman Theatre of Zaragoza (BL) and one from the museum of Málaga (29 MM). They are displayed in Figure 4 and present only one large resonance centered at 12 ppm, suggesting that dolomite is largely predominant in all samples. Unfortunately, the simulation of each spectrum does not give reliable physicochemical parameters in order to use them as provenance markers (C_Q_ around 1.7 MHz, η_Q_ between 0 and 0.3 and a LB/GB ratio of approximately 40/40; see table inserted in Figure 4) owing to a low signal-to-noise ratio.

^25^Mg NMR spectra were also recorded for the same five samples as those analysed for ^43^Ca NMR (three quarry samples and two artifacts): this isotope has a 5/2 spin and a magnetogyric ratio (−1639 × 10^7^ rad s^−1^T^−1^) lower than ^43^Ca. Nevertheless, its 10% natural abundance makes it easier to observe than calcium (Appendix A). ^25^Mg NMR spectra are displayed in Figure 5 and show an exploitable pattern to resolve provenance. Indeed, the ^25^Mg NMR spectrum simulation revealed the existence of two distinct environments in different proportions for Mg that highlight the nonstoichiometric nature of the dolomite studied here [32]. Because the formation of these two phases is linked to the geologic history of dolomite, their quantification can be used as a provenance marker. The results are shown in Figure 5, in which spectra simulation shows a different ratio between the two sites depending on the geographic provenance of the marble. All of the physico-chemical entities obtained are reported in Table 2. Interestingly, ^25^Mg spectra simulation of the two artefacts, 29 MM and BL, have different ratios between the two sites whose proportions allow a provenance attribution. Values for 29 MM are close to those obtained for the dolomitic marble from the quarry of Coín (Mg(1)/Mg(2) 31:64 for 29 MM, similar to the proportion observed for CN1), as viewed by ^13^C NMR, and confirming once again its provenance proposed by Lapuente et al. [4]. For the Zaragoza Roman theatre decoration BL, the ^25^Mg spectrum simulation reveals proportions between the two Mg sites close to those of the Cape-Vathy quarry (64:36 for BL, similar to the proportion observed for THV7). These first results are very encouraging and will require the analysis of more quarry samples in the future.

### 3.2. Elemental Analysis

ICP-OES was applied to quantify eight minor elements present in the samples: Al, B, Ba, Fe, K, Mn, Si, Sr, and Ti, as well as one major element, Mg. These elements were chosen because they have been shown to discriminate between geological locations [10]. The Mg content was also measured to assess its variation. Their mean contents are reported in Table 3 and their analysis, using box plots, in Appendix A. Significant dispersion can be observed within the samples coming from the same quarry. For this reason, provenance identification of artifacts based on the simple observation of elemental quantities is very difficult. Indeed, there is no element whose quantities should be used as a provenance marker, contrarily to the Sr content, which discriminates Carrara from Göktepe marbles [33]. As an exploratory approach, principal component analysis (PCA) was performed on the transformed compositional data (centered log-ratio—CLR; Figure 6A). The use of the CLR transformation prior to PCA arises from the particular nature of the compositional data and the so-called constant sum constraint [29]. The first two axes of the PCA explain 82% (60 and 22%, respectively) of the observed variance. Al content was the main contributor to the construction of the first axis (29%), while iron content was the main contributor to the construction of the second axis (48%). The result of the PCA presented in Figure 6A illustrates the limitation for the assignment of samples because of overlapping, notably between values collected from the Lez-Saint-Béat and Thasos quarries. However, the samples from the two Baetica Province quarries, Coín and Mijas, can be easily distinguished from each other and from the other quarries in this graph, thanks to their discriminant levels of Sr versus Fe, confirming once again the use of Coín marble for 29 MM.

### 3.3. Statistical Analysis

All methodologies taken independently cannot provide provenance proof for most of the artifacts studied. As an additional example, we report in Appendix A a graph displaying all of the δ ^13^C versus d ^18^O isotopic data collected in different publications for the two Malaguese quarries Coín and Mijas [9] and from the Cape-Vathy quarry of Thasos [21], as well as from French Pyrenean quarries found in the district of Lez-Saint-Béat [10]. Three distinct groups are observed related to the provenance of the dolomitic marbles: the Pyrenean marbles are gathered in the left part of the graph (δ ^18^O < −4 PDB (Pee Dee Belemnite)), whereas the Spanish marbles are concentrated in its right part (δ ^18^O > −4 PDB), but without a differentiation between the two quarries of Coín and Mijas; the Thasian dolomitic marble straddles these two groups of values (see Appendix A). When the items’ isotopic values are projected on the graph, those from the museum of Málaga (MM) clearly match with baetician marbles. However, the two Malaguese quarries, Coín and Mijas, can be easily distinguished when elemental content is combined with ^13^C NMR data (Figure 6B). The first two axes of the PCA performed on both compositional data and NMR data explain 71% of the observed variance. Again, Al content was the main contributor to the construction of the first axis (22%), while iron content, NMR intensity, and Δν_1/2_ were the main contributors to the construction of the second axis (29%, 29%, and 27%, respectively). This illustrates the sensitivity of NMR spectra to the presence of paramagnetic elements. PCA results allow the identification of the provenance of two of the items: 6 MM from Coin and 26 MM from Mijas, as previously suggested by Lapuente et al. [9]. The provenance of 29 MM remains ambiguous, and the main difficulty is distinguishing Thasian marbles from Pyrenean marbles as overlapping still persists (Figure 6B).

Thus, PCA was performed by adding isotopic data to compositional and NMR data (Figure 6C). PCA was performed on three samples from the French Pyrenees quarry, seven from Thasos, three from Coín, and three from Mijas, for which all of the data were collected (elemental analysis, ^13^C NMR, and isotopes). The first two axes of the PCA explain 71% (43% and 28%, respectively) of the observed variance. Mg content was the main contributor to the construction of the first axis (20%), while iron content, NMR intensity, and Δν_1/2_ were the main contributors to the construction of the second axis (30%, 29%, and 27%, respectively). Interestingly, the separation between Thasian and Pyrenean marbles is clearly achieved and allows for the identification of the provenance of the marble used to make the decorations of the Roman Theatre of Zaragoza. The PCA suggests that BL is from Thasos and not from the Pyrenees, in accordance with the preliminary results obtained with ^25^Mg NMR.

## 4. Discussion

The main archaeological goal of this work was to shed light upon dolomitic marble trade during Antiquity, and especially the exportation of Thasos dolomitic marble in all of the Mediterranean Basin that persisted for more than a millennium (6th century B.C. to 6th century A.D. [3]), versus more confidential sources. The use of different sources generates historical issues, notably to correlate the marble source to the artwork, its sponsor, or even its use.

For this purpose, the provenances of the marble used to sculpt different artifacts were identified: some decorations of the prestigious Roman Theatre of the antique town of Caesar Augusta (actual Zaragoza, Spain) and statues discovered in the region of Málaga. All of these artifacts date from the 1st to 2nd centuries A.D. and the strategy developed is based on a comprehensive approach that consists of processing a set of analytical data composed of judiciously chosen geologic marbles with respect to the artifacts studied and the artifacts themselves. This strategy allows to add original scientific methods on a obtainable series of samples.

NMR has been selected because of its non-destructive property and its capability to obtain information on the atomic-scale structural environment of different nuclei. Three nuclei were tracked in this study, carbon, calcium, and magnesium, through their respective NMR-sensitive isotope ^13^C, ^43^Ca, and ^25^Mg. Each of them provides complementary information able to help in the decision-making process. ^13^C signal is sensitive to the presence of paramagnetic ions in its vicinity such as iron, manganese, and so on. ^43^Ca NMR spectrum allows to distinguish and quantify dolomite and calcite crystals and ^25^Mg is sensitive to structural disorder, i.e., imperfections in the crystalline lattice, that necessarily occurs because of the non-stoichiometric nature of dolomite rocks. The latter nucleus presents an interesting performance for provenance discrimination owing to its ability to quantify the two distinct Mg environment of dolomitic marbles.

A common problem in the determination of the provenance of marbles is the fact that the analytical data collected by most authors have been applied to different collections of samples, hindering a global multivariate analysis. To date, no single technique is able to determine the provenance of a marble. To overcome these difficulties, it is recommended, first, to use different and complementary methods in terms of collected information (presence of specific trace elements or isotopes, macroscopic texture, molecular organization, and so on) and, second, to acquire all the data on the same collection of samples in order to proceed to the statistical data analysis for determining reliable provenance. This is why we collected NMR and ICP-OES data on samples already analyzed by petrography and cathodoluminescence by one of us, and whose isotopic ratios were already measured (see Appendix A and references herein). When applied to a single method, for most of the studied artifacts, the provenance was not determined, but when all of the data collected were used, the groups of points representative of a quarry are clearly separated, leading to a provenance determination. 

Thus, the artifacts 6 MM, which is part of a triumphal arch commemorating the victory over the Maurii (176 AD), and 29 MM, a male figure believed to be a distinguished person wearing a mantle, and found in Cartima, Málaga province dating from the first half of the 2nd century AD, are made of local dolomitic marble, and more particularly from the quarry of Coín. 24 MM, a fountain-sculpture from the villa de el Secretario, found at Fuengirola, Málaga province and dating from the mid-2nd century AD, is made of dolomitic marble extracted from the Mijas quarry. Were the two quarries exploited at different times, or were these quarries exploited for different purposes and/or clients depending on their social hierarchy? Regardless, in all these cases, the local marble source was preferred.

The decoration of the Roman Theatre of Caesar Augusta, BL, has been identified as a Thasian marble. In this case, the farthest marble was favored: is it related to transport conditions—the sea route being easier than the land one? Or is it linked to the prestige of the building? We will let historians answer these questions!

## Figures and Tables

**Figure 1 materials-16-01468-f001:**
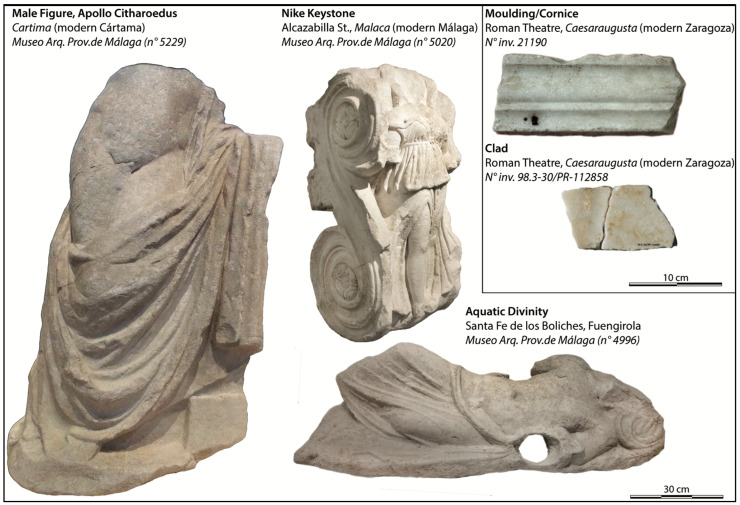
Artifacts studied in this work. Male Figure (29 MM), Nike Keystone (6 MM), Aquatic Divinity (26 MM) from the Málaga museum (credit Málaga museum). Moulding (CL) and Clad (BL) from the Roman Theatre of Zaragoza (credit, Roman Theatre).

**Figure 2 materials-16-01468-f002:**
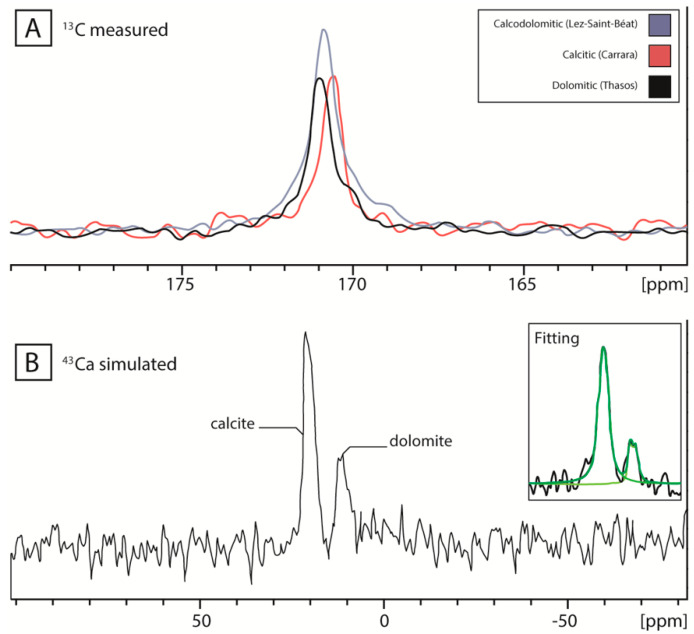
(**A**) ^13^C NMR spectra of dolomitic (Thasos, black), calcitic (Carrara, red), and dolomito-calcitic (Lez-Saint-Béat, written Saint-Béat in the figure, blue) marbles. (**B**). ^43^Ca NMR spectrum of dolomito-calcitic (Lez-Saint-Béat). Insert represents the best fits (in green) obtained for ^43^Ca NMR spectrum simulation using the following physico-chemical parameters: calcite, δ_iso_ 22.7 ppm, C_Q_, 1.4 MHz, η_Q_ 0.1; dolomite, δ_iso_ 14.0 ppm, C_Q_, 2.0 MHz, η_Q_ 0.5.

**Figure 3 materials-16-01468-f003:**
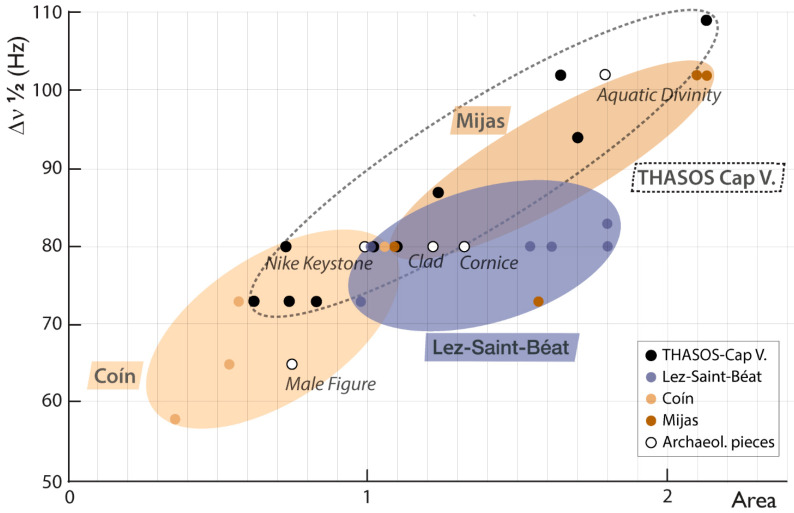
Area vs. full width at half-maximum (Δν_1/2_ in Hz) of ^13^C NMR resonances obtained for different dolomitic marbles, and first items’ provenance assignment (Male Figure 29 MM).

**Figure 4 materials-16-01468-f004:**
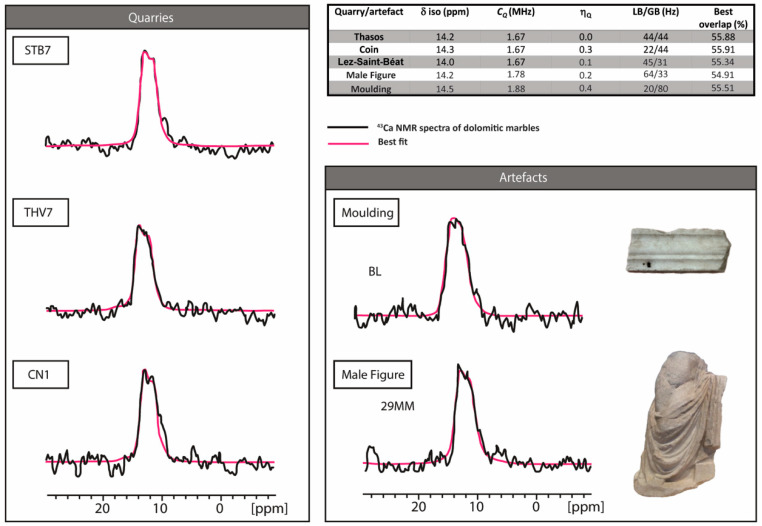
^43^Ca NMR spectra of dolomitic marbles (black) and their best fit (sola software Bruker Biospin, pink) using the parameters displayed in the table. STB7: Lez-Saint-Béat quarry, THV7: Thasos Cap Vathy-Quarry, CN1, Coin Quarry, 29 MM, male figure from the Museum of Malaga, BL, molding from the Caesar Theatre of Zaragoza.

**Figure 5 materials-16-01468-f005:**
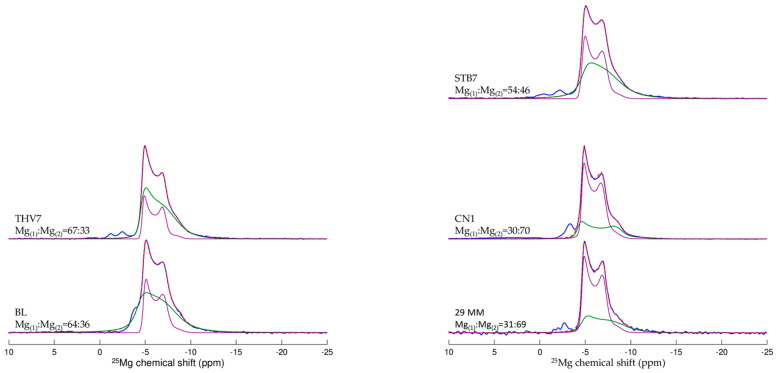
^25^Mg NMR spectra of dolomitic marbles (blue) and their best fit using the parameters displayed in Table 2. STB7: Lez-Saint-Béat quarry, THV7: Thasos Cap-Vathy Quarry, CN1, Coin Quarry, 29 MM, male figure from the Museum of Malaga, BL, moulding from the Roman Theatre of Zaragoza.

**Figure 6 materials-16-01468-f006:**
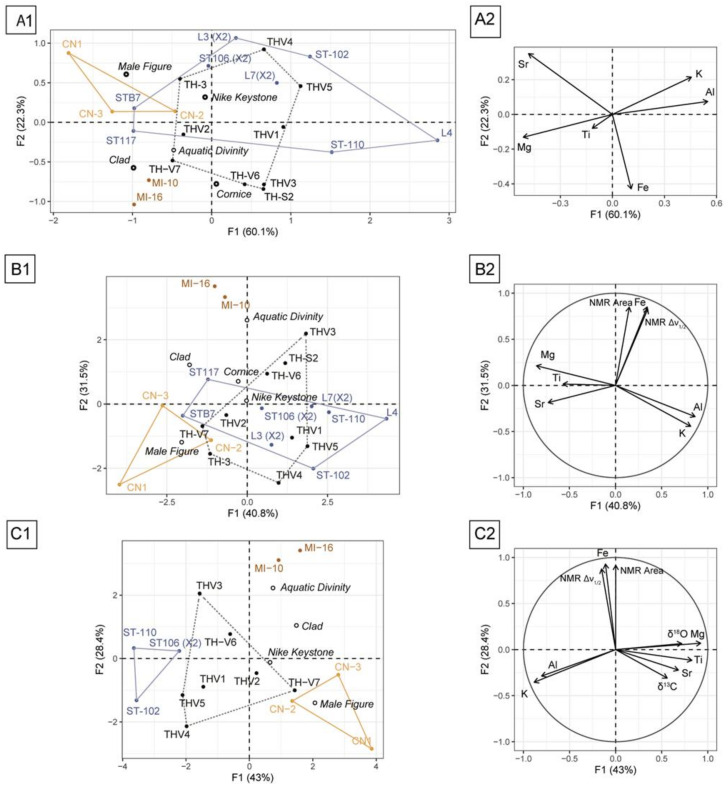
Principal component analysis (PCA) of marble data. (**A**) Non-standardized PCA of log-transformed ICP-OES data. (**B**) Standardized PCA of log-transformed ICP-OES data and ^13^C NMR. (**C**) Standardized PCA of log-transformed ICP-OES data, ^13^C NMR, and isotopic data. (1) Individuals factor map. (2) Variables factor map.

**Table 1 materials-16-01468-t001:** Physico-chemical parameters extracted from ^13^C NMR spectra of quarries and artefacts. δ corresponds to the NMR ^13^C chemical shift with respect to tetramethyl silane, which resonates at 0 ppm; area is the integration of the area of the signal with respect to an external reference; Δν_1/2_ is the full width at half-maximum of the carbonate resonance. All of the values are expressed as average ± STD. Details are reported on Appendix A.

Quarry/Artifacts	δ (ppm)	Area	Δν_1/2_ (Hz)
Coín (n = 4)	170.80 ± 0.03	0.6 ± 0.3	69 ± 9
Mijas (n = 4)	170.82 ± 0.01	1.7 ± 0.5	89 ± 15
Lez-St-Béat (n = 6)	170.8 ± 0.3	1.5 ± 0.7	79 ± 3
Thasos (n = 7)	170.80 ± 0.02	1.3 ± 0.5	85 ± 13
**6 MM**	170.81	1.0	80
**24 MM**	170.79	1.8	102
**29 MM**	170.81	0.7	65
**BL**	170.79	1.2	80
**CL**	179.76	1.3	80

**Table 2 materials-16-01468-t002:** Physico-chemical entities extracted from ^25^Mg NMR spectra fit. Ratio expresses the proportion of the two sites defined to fit the NMR spectrum (in %); isotropic chemical shift δ_iso_ (ppm), distribution of isotropic chemical shift Δδ_iso_ (ppm), average quadrupolar coupling constant C_Q_ (MHz), average asymmetry parameter η_Q,_ distribution of quadrupolar coupling constant ΔC_Q_ (MHz), and “disorder weight” ε.

Site	Ratio (%)	δ_iso_	Δδ_iso_	C_Q_	η_Q_	ΔC_Q_	ε
**29 MM**							
Mg_(1)_	31 ± 3	−3.80 ± 0.04	0.74 ± 0.10	1.13 ± 0.02	0.15	0.30 ± 0.04	0.12
Mg_(2i)_	69 ± 3	−4.00 ± 0.00	0.13 ± 0.01	0.90 ± 0.00	0.25	0.09 ± 0.00	0.12
**BL**							
Mg_(1)_	64 ± 1	−3.50 ± 0.04	1.37 ± 0.06	1.05 ± 0.06	0.15	0.30 ± 0.01	0.12
Mg_(2i)_	36 ± 1	−4.20 ± 0.01	−0.32 ± 0.02	0.89 ± 0.02	0.25	0.11 ± 0.00	0.12
**CN1**							
Mg_(1)_	30 ± 1	−3.10 ± 0.04	0.46 ± 0.05	1.20 ± 0.05	0.15	0.13 ± 0.02	0.12
Mg_(2i)_	70 ± 1	−4.00 ± 0.01	0.14 ± 0.01	0.88 ± 0.01	0.25	0.10 ± 0.00	0.12
**STB7**							
Mg_(1)_	54 ± 1	−4.10 ± 0.02	1.35 ± 0.05	1.02 ± 0.05	0.15	0.33 ± 0.01	0.12
Mg_(2i)_	46 ± 1	−4.10 ± 0.00	−0.46 ± 0.01	0.90 ± 0.01	0.25	0.10 ± 0.00	0.12
**THV7**							
Mg_(1)_	67 ± 1	−3.90 ± 0.01	0.37 ± 0.01	1.02 ± 0.01	0.15	0.35 ± 0.00	0.12
Mg_(2i)_	33 ± 1	−4.10 ± 0.00	−0.21 ± 0.01	0.89 ± 0.01	0.25	0.08 ± 0.00	0.12

**Table 3 materials-16-01468-t003:** Elemental analysis of dolomitic marbles from the different quarries studied. Values are expressed ± standard deviation in ppm, except for Mg (%, *w/w*). * nm: not measurable (value under the threshold, 5–10 ppm).

Quarry/Element	Lez-St-Beat	Coín	Mijas	Thasos	6 MM	24 MM	29 MM	CL	BLN
**Al**	67 ± 28	70 ± 75	300 ± 400	190 ± 120	159	106	52	102	24
**B**	11 ± 1	11 ± 1	7 ± 6	10 ± 1	10	10	10	10	nm
**Ba**	4 ± 6	nm*	5 ± 8	nm	nm	nm	nm	nm	nm
**Fe**	161 ± 40	69 ± 51	262 ± 120	178 ± 88	150	321	76	138	105
**K**	160 ± 100	40 ± 15	230 ± 320	158 ± 105	129	89	64	61	37
**Mn**	90 ± 20	16 ± 4	44 ± 16	19 ± 17	17	50	18	30	32
**Si**	4 ± 6	10 ± 9	24 ± 3	6 ± 8	16	20	nm	18	nm
**Sr**	89 ± 17	62 ± 16	14 ± 5	40 ± 27	74	79	92	16	25
**Ti**	18 ± 4	18 ± 3	18 ± 2	21 ± 4	25	20	17	20	14
**Mg (%)**	8.4 ± 1.1	8.8 ± 1.8	10.1 ± 0.9	8.9 ± 0.6	9.8	11.6	9.0	10.3	7.5

## Data Availability

Not applicable.

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
