# Peer review of "13C, 25Mg, and 43Ca Solid-State NMR for the Purpose of Dolomitic Marbles Provenance Elucidation"

_materials, 2023, doi:10.3390/ma16041468_

Round 1

Author Response

Dear reviewer, see below the answer of all your questions and remarks:

Abstract.

A specific sentence concerning the efficiency 25Mg NMR for dolomitic marble provenance has been added.

L17-18 : the term “origin” has been changed by provenance, as proposed, in all the text

L19: cathodoluminescence and elementary analysis are replaced by cathodoluminoscopy and elemental analysis.

L21: deciphering is replaced by characterization

L22: I don’t agree with the referee. We look at Nuclei, but the isotope sensitive to NMR is the 13C nuclides.

L26 : raw materials has been changed by dolomitic marbles

Keywords. I think they are the good one

Introduction.

L 30, singular has been removed

L34, Dolomieu would refer to the marbles traditionally used by artists to sculpt objects, see ref 1.

L35 : classical counterpart has been changed by calcitic marbles.

L39: species has been changed by varieties

L40-42 : has been changed as proposed by the referee

L46-48 : references has been added

L53 to 61 has been changed.

L74 the two strategies have been re-explained for more clarity

Material and methods

Modifications are done in order to avoid repetitions as requested.

L97-103 has been deleted

L107-137 : The NMR technique has been already described by the authors for marble provenancing, and the present article refers to them (ref 15, 16 and 34)

L141-147 : the protocole has been described in ref 16 as mentioned in this part

L143: “recipient” has been replaced by “container”

L144: “disrupted is replaced by dissolved

L146-147: has been explained in the text

Results

L181: 30 mg is sent for this analysis, but a smaller amount is used for the analysis

Table 1. the significance of these parameters has been already described in ref 16. For the small number of samples, each experiment is long (3 to 4 h for One sample, 2/day …) and machine time shared with over 200 researchers! But, fortunately their number increases with time. Precisions are added in the caption and all the data reported in supplementary materials (S3). The values between Table 1 and figure 3 has been checked.

 L184-202, separation has been made between results and discussion when necessary.

  1. 184, The NMR theory can be find in the web, and explanations are done for the observable and not observable nuclides depending on their spin.

L185. Paramagnetic elements has been added

L204. Has been changed in S2 (0.135%)

L206 : precision has been done

L 212 : calcitic has been added

L210-215: Is reported for the first time the 43Ca NMR characteristic of dolomite. Precision is added in the text.

Figure 2. ppm is the currently scale used for NMR, referred to a reference at 0 ppm (for 43Ca, a solution of CaCl2 1M is used to determine the 0)

L222-228 . We don’t agree with the referee, for us the calcite/dolomite ratio measured is a result.

L225-226. For 2 molecules of carbonate, the number of Calcium atom is 2 for calcite (CaCO3) and 1 for dolomite (CaMg(CO3)2). It is now mentioned in the text.

L226-228. The sentence has been changed  and Mont-Rié removed (this is the name of the mountain near Saint-Béat)

Figure 3. has been changed with the modifications requested.

L233. Referee is write, the sentence has been changed by taking into account the fact that no samples from Mijas was studied by Ca or Mg NMR

Figure 5 has been added, this was an oversight.

L244-245. Instruments for measuring Mg spectra are not numerous, and are subject to specific demands, with long lead times. This is why we only present the preliminary results.

L245-251. Precisions are made to help the reader to understand the two different environments leading to different behaviors of the Mg in dolomite.

L247. We don’t think that XRD will help to see the environment of Mg in the crystal of dolomite.

L253-254it is a %, added in the text

L259. “a larger corpus “ has been replaced by “the analysis of more quarry samples”

Figure 4. The different Nuclides observed have different spin values, and different physico-chemical parameters. 13C has a spin ½ with no quadrupolar moment contrarily to 25Mg or 43Ca, this phenomenon explains the different parameters observed from one table to another.

Table 2. Explanations are in the text for the two Mg “sites” (so with different environments, and thus different physico-chemical parameters), they are expressed as % (20% for Mg1, 80% for Mg2 …

L267-268 : “minor” has been removed

L270. The sentence has been changed to take into account the referee comment.

Figure 6 has been changed (the transformation of the image into a pdf file has removed crucial information)

L300-321. Corrections have been made with respect to normalized standards.

L328- 347: we don’t agree with the referee, these parts belong to the results part, since it gives partial conclusions.

L354 : … means etc (it has been changed in the text

L356 :  “structural disorder” means imperfections in the crystalline lattice, that necessarily occurs due to the non-stoichiometric nature of dolomite rock. It has been added in the text

L367-369: petrographic data, MGS, cathodofacies were previously reported in ref 9 and 10, and were taken as the basis of this work.

L374-387: we don’t take into account other possible dolomitic marbles, because these provenances were eliminated due to results obtained in previous works (see ref 9 and 10)

Supplementary materials have been modified as required.

Reviewer 2 Report

The authors have submitted a manuscript describing the application of solid-state NMR spectroscopy to characterise the provenance of marbles. They have collected a range of NMR data and applied statistical methods to analyse them.

Unfortunately, however, the manuscript as such is in a bad shape. It contains lots of typographical and language errors, to the extent that the readability of the text really suffers. One example is the description of the „phase-out strategy“ in the Introduction (page 2). The sentence on lines 59-61 does not make sense, something in the vocabulary and/or the grammar went wrong. Also, the text contains numerous inconsistencies, for example the random use of capitalization (figure/Figure, Iron/iron…). There are frequent typos like „Satistical softwares“ (pg. 3), „Stastitical analysis“ (pg. 11) and crude translation failures such as the „Zaragoza antic Theatre“ (pg. 8). The authors should google the difference between „antic“ and „antique“’! The most blatant of all these mistakes and oversights are certainly the text parts on page 2/3 which have been just left in from the MDPI template — this neatly demonstrates how little effort the authors (all seven of them, which are also not properly separated by commas on the title page) have invested in this manuscript. This paper really needs a thorough revision before a reviewer can be asked to spend time on it.

Author Response

Dear reviewer, 

please find below the answer to your questions, remarks :

- L59-61: the phase-out strategy has been re-explained in the text, and the sentence has been changed

- Typos are corrected p2 and 11, as capitalization harmonized all along the text

-antic has been replaced by antique or Roman

- L97-103 has been removed.

Reviewer 3 Report

The present manuscript from Pianet et al. is an interesting description on how the combination of NMR, elemental analysis and isotopic abundance analysis can be used to establish the provenance of marble used in ancient and historical artifacts.

The NMR results are especially interesting and useful to support the reported conclusion and the subject on this manuscript is interesting and adequate to the Materials journal.

Nevertheless, several results are exposed in a not clear and confused manner, at the level that it is not possible to evaluate if the conclusions are well supported or not.

One of the most important figures, Figure 5, which should report the results of the PCA analysis, is totally unreadable! No indication of the axes labels, no legend, none indication of the color-code used,…. it is not possible to understand what is what in this figure.

Figure 3 also is reported with no indication of the color code used for each spot in the figure…making it unclear.

In figure 2 legend is reported in French…..!?!

In the tables numbers are reported with commas or points …or a mixture of them.

In table 2 it is not clear why some numbers are colored or partially colored.

An entire paragraph form page 2 line 97 to page 3 line 104 has no sense with the present discussion and it should be removed. It looks like it was present in a previous template, and it was not removed!

The present manuscript cannot be accepted in the present version, it has to be fully revised and resubmitted with clearer figures and recorrected tables and text.

Author Response

Dear reviewer,

please find below the answer to your comments,

Figure 6 (5 was missing) has been changed (the transformation of the image into a pdf file has removed crucial information).

Figure 3 also is reported with no indication of the color code used for each spot in the figure…making it unclear. : the missing information has been added

In figure 2 legend is reported in French…..!?! : it has been changed

In the tables numbers are reported with commas or points …or a mixture of them. Corrected

In table 2 it is not clear why some numbers are colored or partially colored. The colors (which indicated the provenance of the considered artifact) have been removed in the new version.

An entire paragraph form page 2 line 97 to page 3 line 104 has no sense with the present discussion and it should be removed. It looks like it was present in a previous template, and it was not removed!: It has been removed.

Round 2

Reviewer 1 Report

The authors have improved the manuscript significantly, but still insufficiently.

The following comments from my previous review were neither addressed nor answered by the authors, so they are sustained.

Abstract

Some important information, which normally appear in abstract, are missing. Where does the material come from? Provide general information about the quarries and artifacts. What are the main direct outcomes of the work? Provide the source of artifacts identified.

It is not really true that the historical significance of the results is discussed in the manuscript. It is only briefly mentioned and could be evaluated in much more detail.

Introduction

The main goals of the work are vaguely expressed. The reader can find them much precisely defined in the following chapters no. 3 “Materials” (l. 154-157, 166-170) and no. 4 “Discussion” (l. 334-339). It is crucial to formulate the goals precisely in the Introduction.

Results

Entire text on p. 4 is misplaced. It belongs either to Introduction or Material and Methods chapters (see above).

Table 1. The number of samples for the quarry samples is too low (from 4 to 7) to report the data by statistical parameters. All results should be provided for all samples measured.

l. 245-251: Authors refer to two “environments”, “phases”, and “sites” in this part, but it is not clear what exactly these terms mean. Do they indicate the same? And what are they? Dolomite and calcite?

Figure 4: Parameters in the table are not explained.

Table 3: What are the values reported for quarry samples? Average? Median value? Number of samples analyzed is missing. If it is the same as in Table 1, then all values for all samples must be given.

l. 356-357: Dolomitic rocks cannot be stoichiometric or non-stoichiometric. It refers to minerals.

Figure S1a: Incorrect caption. It is the content of these elements that is displayed, not composition.

Figure S1b: Whiskers are missing for Coin and Mijas.

Some answers or corrections are still not satisfactory. I provide additional comments (green).

Material and Methods

Description of material and methods is insufficient here. In the following parts of the manuscript some information regarding material is repeated, but additional information appears as well in chapters no. 3 “Results” (l. 157-166), no. 4 “Discussion” (l. 341-347), and Figure 1. The material is referred to in different parts of the manuscript, figures, and tables in different ways, which brings much confusion. Please unify the terminology and use the same terms throughout the manuscript.

There is improvement, but problems still exist. For instance: Saint-Beat is used in Tab. 1, Fig. 2, Tab. S3; Lez is used in Fig. 3, Tab. 3, Tab. S1, Fig. S1; Lez-Saint-Beat is used in Fig. 4, Fig. 5, Fig. S2. Moreover, Coin and Mijas are indicated separately in almost all figures and tables except Fig. S2, where “Coin-Mijas (Malaga)” is used. These inconsistencies must be eliminated, because they bring confusion. I do not have time to check it in more detail. It should be the care of authors to check the manuscript in detail to track and fix these problems throughout.

l. 247: Issues related to stoichiometry of carbonate minerals should be analyzed with the use of XRD.

The authors replied “We don’t think that XRD will help to see the environment of Mg in the crystal of dolomite.” I still do not understand what the authors mean by “environment”, because they did not react to my comment to lines 245-251. Nevertheless, XRD is absolutely the right method to check the stoichiometry of dolomite, because non-stoichiometric dolomite has the main XRD peak d(104) displaced toward higher or lower angles from the position of stoichiometric dolomite. See for example: Reeder, R.J., 1983. Crystal chemistry of the rhombohedral carbonates. Rev. Mineral. 11, 1–47. 

l. 267-268: 8 or 9 minor elements? Inconsistent.

The inconsistency in this sentence is that it is written that 8 minor element were quantified, whilst nine are listed.

l. 356: What do the authors mean by “structural disorder”?

Non-stoichiometric nature of dolomite is not necessarily associated with defects in the crystalline lattice.

l. 367-369: It would be important to know the general fabric, MGS, cathodofacies etc. and to cite previous works reporting these data.

l. 374-387: I suggest to tone down the conclusions. The authors did not take into account other possible dolomitic marble sources (Ephesos, Proconnesos, Sivec), so they should change “is” to “can be” when they indicate the source of artifacts.

If these sources were eliminated by previous workers, this should be clearly stated in the text and relevant references should be cited.

Figure S2: Incorrect caption – it should be something like “Stable C and O isotope composition…” or “δ13C vs δ18O diagram…”. One of the axes is incorrectly described, because it should be δ18O. Horizontal scale should be shortened to range from -9 to 0, whereas vertical scale to range from 0 to 5.

Author Response

Answers to referee 1 :

  • Homogeneization of the quarries name has been done
  • “environment” is classically used by NMR spectroscopists to describe all kinds of external elements susceptible to modify the NMR spectrum profile of a nucleus. These environments modifications can be tracked by NMR but not by XRD. See ref Topical Reviews by Bryce, 2017, 4, 350-359 as an example.
  • L267-268 : there are 9 elements of which 1 (Mg) can’t be considered as minor in dolomitic marbles.
  • L 367-369: the references are all cited in table S1, it has been added in the text.
  • L374-376 : the authors have been cited in the different tables, figures, and in the text on numerous occasions (Lapuente, Lapuente et al, Blanc & Lapuente, Lazzarini …), and to cite them again in this part of the manuscript will be redundant.
  • Figure S2 : has been changed as required

Reviewer 3 Report

The revised version the manuscript now has strongly improved in clarity of the text and quality of the figures. It is now acceptable in Materials.

I have just a couple of corrections:

In the caption of Figure 5 the experimental spectra are blue instead of “black”.

In the figure S2, the plot has the same label in both axes, while one the axes should be 18O!

Please correct them

Author Response

Dear Reviewer,

thank you for your comments, the corrections requested have been made.

best regards

Isabelle Pianet

best regards